# Many, but not all, lineage-specific genes can be explained by homology detection failure

Caroline M. Weisman[1], Andrew W. Murray[1], Sean R. Eddy[1,2,3]*

**1** Department of Molecular & Cellular Biology, Harvard University, Cambridge, Massachusetts, United States of America, **2** Howard Hughes Medical Institute, Harvard University, Cambridge, Massachusetts, United States of America, **3** John A. Paulson School of Engineering and Applied Sciences, Harvard University, Cambridge, Massachusetts, United States of America

* seaneddy@fas.harvard.edu

**Data Availability Statement:** All data used in these analyses and the scripts necessary to reproduce them are available in the Supporting information and on our code repository at http://www.github.com/caraweisman/abSENSE.

## Abstract

Genes for which homologs can be detected only in a limited group of evolutionarily related species, called "lineage-specific genes," are pervasive: Essentially every lineage has them, and they often comprise a sizable fraction of the group's total genes. Lineage-specific genes are often interpreted as "novel" genes, representing genetic novelty born anew within that lineage. Here, we develop a simple method to test an alternative null hypothesis: that lineage-specific genes do have homologs outside of the lineage that, even while evolving at a constant rate in a novelty-free manner, have merely become undetectable by search algorithms used to infer homology. We show that this null hypothesis is sufficient to explain the lack of detected homologs of a large number of lineage-specific genes in fungi and insects. However, we also find that a minority of lineage-specific genes in both clades are not well explained by this novelty-free model. The method provides a simple way of identifying which lineage-specific genes call for special explanations beyond homology detection failure, highlighting them as interesting candidates for further study.

## Introduction

Homologs are genes that descend from a common evolutionary origin. "Lineage-specific genes" are defined operationally as genes that lack detectable homologs in all species outside of a monophyletic group [1]. Also referred to as "taxonomically-restricted genes" [2, 3], and as "orphan genes" when found only in a single species [4, 5], they are ubiquitous in the genomes of sequenced organisms. For example, by previous reports, 23% of *Caenorhabditis elegans* genes are specific to the *Caenorhabditis* genus [6]; 6% of honey bee genes are specific to insects [7]; 25% of ash tree genes are specific to the species [8]; and 1% of human genes are specific to primates [9].

Where do lineage-specific genes come from? A common interpretation is that they are "novel" genes. Various proposals for the molecular nature of this novelty have been advanced. For example, lineage-specific genes have been interpreted as having evolved from previously noncoding sequence ("de novo originated" genes) [5, 10, 11] and as duplicated genes that have gained a novel function, diverging radically and beyond recognition from their homologs in

**Funding:** This work was primarily funded by a Howard Hughes Medical Institute investigator award to SRE. SRE is also supported in part by NIH (R01-HG009116) and AWM is supported in part by grants from the NIH (R01-GM43987), and the NSF-Simons Center for the Mathematical and Statistical Analysis of Biology (NSF #1764269, Simons #594596). Computations were done on the Cannon cluster supported by the FAS Division of Science, Research Computing Group at Harvard University. The funders had no role in study design, data collection and analysis, decision to publish, or preparation of the manuscript.

**Competing interests:** The authors declare that no competing interests exist.

**Abbreviations:** BUSCO, Benchmarking Universal Single Copy Ortholog; Mya, million years ago; ORF, open reading frame; YGOB, Yeast Gene Order Browser.

the process ("neofunctionalized" genes) [5, 12]. Though different in detail, these proposals share the key assumption that a lack of detectable homologs indicates some kind of biological novelty: Lineage-specific genes either have no evolutionary homologs or no longer perform the same function as their homologs outside the lineage [13–16]. We refer to these interpretations collectively as the "novelty hypothesis" of lineage-specific genes. The novelty hypothesis has informed work on the evolution of new features at molecular, cellular, and organismal scales [16–20].

An alternative explanation for a lineage-specific gene is that nothing particularly special has happened in the gene's evolutionary history. The gene has homologs outside of the lineage (no de novo origination), and no novel function has emerged (no neofunctionalization), but despite this lack of novelty, computational similarity searches (e.g., BLAST) have failed to detect the out-of-lineage homologs. We refer to such unsuccessful searches as homology detection failure. As homologs diverge in sequence from one another, the statistical significance of their similarity declines. Over evolutionary time, with a constant rate of sequence evolution, the degree of similarity may fall below the chosen significance threshold, resulting in a failure to detect the homolog. Some lineage-specific genes may just be those for which this happens to have occurred relatively quickly, even in the absence of any novelty-generating evolutionary mechanisms.

The possibility of homology detection failure has long been recognized, but the key questions of how many and which lineage-specific genes are best explained by it remains unclear. Previous work has aimed to explicitly simulate the evolution of each gene. These approaches depend on many evolutionary parameters, to which results have proven sensitive, ranging widely within the same taxon [21–27]. A recent study elegantly avoids this problem, estimating the overall rate of detection failure from a subset of genes for which direct comparison of syntenic orthologous coding sequences is possible, allowing highly diverged orthologs to be identified. Extrapolating this rate genome-wide, it proposes that around 40% and 25% of lineage-specific genes in fungal and fly lineages are due to detection failure [28]. This estimate relies on this subset of genes being a representative sample. Additionally, the underlying method cannot assess whether a particular lineage-specific gene is due to detection failure unless the syntenic orthologous region is identifiable, which is typically not the case.

Here, we describe a simple method for evaluating whether homology detection failure is sufficient to account for a particular lineage-specific gene. We develop a mathematical model that estimates the probability that a homolog would be detected at a specified evolutionary distance if it was evolving at a constant rate under standard, novelty-free evolutionary processes. The resulting method can be used on any gene with detected homologs in at least 2 taxa. We apply the method to all such lineage-specific genes in insects and yeasts and find that many, but not all, lineage-specific genes in these taxa can be explained by homology detection failure. This method should be easily applicable to other taxa, where it can be used to determine which lineage-specific genes are unlikely to be due to detection failure, highlighting them as candidates for true evolutionary novelty.

## Results

### A null model of homolog detectability decline as a function of evolutionary distance

We developed a formal test of the null hypothesis that homology detection failure is sufficient to explain the lineage specificity of a gene. Specifically, we model the scenario in which the gene actually existed in a deeper common ancestor, evolved at a constant rate, and has homologs outside the clade in which its homologs are detected that appear to be absent solely due to

homology detection failure. This is an evolutionary null model: It invokes no processes beyond the simple scenario of orthologs diverging from a common ancestor and evolving at a constant rate.

Because of its use in previous work on lineage-specific genes and in sequence analysis more broadly, we use BLASTP (available from the National Center for Biotechnology Information, https://blast.ncbi.nlm.nih.gov/Blast.cgi) as the search program used to detect homologs here. In search programs like BLAST, sequence similarity is used to infer homology between 2 genes. Such programs report a similarity score (referred to as "bitscore" by BLAST) between a pair of sequences, as well as the number of sequences that would be expected to achieve that similarity score by chance (an E-value). When this number falls below a significance threshold (e.g., E < 0.001), statistically significant similarity is interpreted as evidence that the 2 genes are homologous. The similarity score therefore directly determines whether a homolog is successfully detected in a search.

The key idea in our method is to predict how the similarity score between 2 homologs evolving according to our null model is expected to decline as a function of the evolutionary distance between them. We can then ask whether a given gene's lack of detectable homologs outside of the lineage is expected under this null evolutionary model.

We analytically modeled how the similarity score between 2 homologs decays with the evolutionary distance between them. A simple argument shows that similarity scores are expected to decline roughly exponentially with evolutionary time. Suppose we assume that the similarity score between 2 homologs is proportional to the percent identity between them and that every position in the protein changes at the same protein-specific rate, which is constant over evolutionary time. Then the expected similarity score $S$ between 2 homologs separated by an evolutionary divergence time $t$ is given by $S(t) = Le^{-Rt}$, and the variance of this similarity score is given by $\sigma^2 = L(1 - e^{-Rt})(e^{-Rt})$ (S1 Supplemental Information). The protein-specific parameter $L$ is related to the protein's length. The protein-specific parameter $R$ is related to the rate at which the protein accumulates substitutions over evolution and so incorporates protein-specific factors that contribute to that rate, including mutation rate at the locus and the effects of selection on the protein.

Although an exact derivation of this function assumes a substitution-only process with a single constant position-independent substitution rate, the same functional form will approximate the effects of site-specific rates (S1 Supplemental Information) and position-specific insertion/deletion. Whatever the detailed position-specific selection pressure on the protein, if it is constant over time, we expect similarity scores to decline roughly exponentially. We can empirically estimate this exponential by fitting $L$ and $R$ to observed scores at different divergence times. By subsuming the complex effects of selective pressure into a 2-parameter empirical model of similarity score decline, we avoid the need for a parameter-heavy model of sequence evolution. Minimizing the number of parameters in the model allows us to apply it to genes with a limited number of identified homologs (genes specific to very young lineages) while minimizing the problems of results being sensitive to parameter estimation that occur in complex models. The assumption that rate $R$ is the same across evolutionary time and in all lineages is also clearly a simplification, but this is the null hypothesis that we aim to test: We aim to identify genes in which a lack of detected homologs is consistent with an expected constant decay of similarity scores with time, without any need to invoke lineage-specific appearance or rate shifts.

We can predict similarity scores for a given gene if we have 3 inputs: a gene from a chosen focal species, the similarity scores of successfully identified homologs of the gene in a at least 2 other taxa *(S)*, and the relative evolutionary distances between the focal species and these other species *(t)*. (As described in the following section and Methods, we precalculate these

evolutionary distances $t$ from an aggregate of many genes from the set of species under consideration, and therefore, they do not depend on the particular gene under consideration.) We use these inputs to find the gene-specific values of the parameters $L$ and $R$ that produce the best fit to our equation describing how similarity scores decline within the species in which homologs were detected. We then use these parameters to extrapolate and predict the expected similarity score of hypothetical homologs of the gene at evolutionary distances beyond those of the species whose homologs were used in the parameter fitting. Given an E-value threshold, this predicted similarity score, and the expected variance of the similarity score, we can estimate the probability that a homolog will be undetected at these longer evolutionary distances. In the analyses that follow, we use a relatively permissive E-value threshold of 0.001.

This key idea is illustrated in Fig 1, which shows examples of fitting similarity scores versus evolutionary distance for several different yeast and insect genes.

## The null model adequately describes the decay of ortholog detectability with evolutionary distance

We applied our model to genes of the yeast *S. cerevisiae* and the fly *D. melanogaster* and their orthologs in several fungal and insect outgroups, respectively. We focus on the fungi and insects because their genomes are well-annotated, they have closely related and well-annotated sister species, and they have been the focus of previous work on lineage-specific genes [5, 11, 29–32]. For *S. cerevisiae*, we included 11 fungal species spanning a divergence time of approximately 600 million years [33, 34]; for *D. melanogaster*, we included 21 insect species spanning a divergence time of approximately 400 million years [35]. These species are listed in Fig 2.

Before using our null model to ask whether it explains the lack of detected homologs of lineage-specific genes, we confirmed that it is a good approximation of how similarity scores decay with evolutionary distance. To do this, we tested how well the model represents the decay of similarity scores of general *S. cerevisiae* and *D. melanogaster* genes in increasingly distant species. If the model fits this decay well for most genes, it is likely a good representation of the minimal evolutionary process in the null hypothesis and can therefore detect deviations from that process.

To obtain evolutionary distances from the focal species ($t$ values), represented by the x-axis in Fig 1, we used 102 genes from the Benchmarking Universal Single Copy Ortholog ("BUSCO") [36] database to calculate evolutionary distances in substitutions/site between *S. cerevisiae* and each of the 11 other fungi and 125 BUSCO genes to calculate evolutionary distances between *D. melanogaster* and each of the 21 other insects (Methods). We note that, because this approach directly calculates the pairwise distance in substitutions/site between the focal species and each other species, it incorporates changes in evolutionary rates across lineages without assuming a lineage-invariant molecular clock. In both taxa, to show that distances can be reliably computed using a small number of genes, we also re-calculated these distances using 2 random subsets of 15 BUSCO genes. Distances computed from these different gene sets were similar (S1 Table). Fig 2 shows evolutionary distances inferred from one of the 15 gene sets between the focal organism *S. cerevisiae* and the 11 other fungi and between the focal organism *D. melanogaster* and the 21 other insects. For reference, Fig 2 depicts these distances along with a topology taken from previous phylogenetic studies of these taxa [34, 35]; branch lengths are not to scale. We use these distances, computed from one of the 15 gene subsets, in all results presented in the main text beyond this point.

We next took all annotated *S. cerevisiae* and *D. melanogaster* proteins (S2 Table) and identified the similarity scores of their detectable orthologs in each of the 11 other fungal and 21 other insect outgroup species, respectively. (For *S. cerevisiae* and *D. melanogaster*, the score is

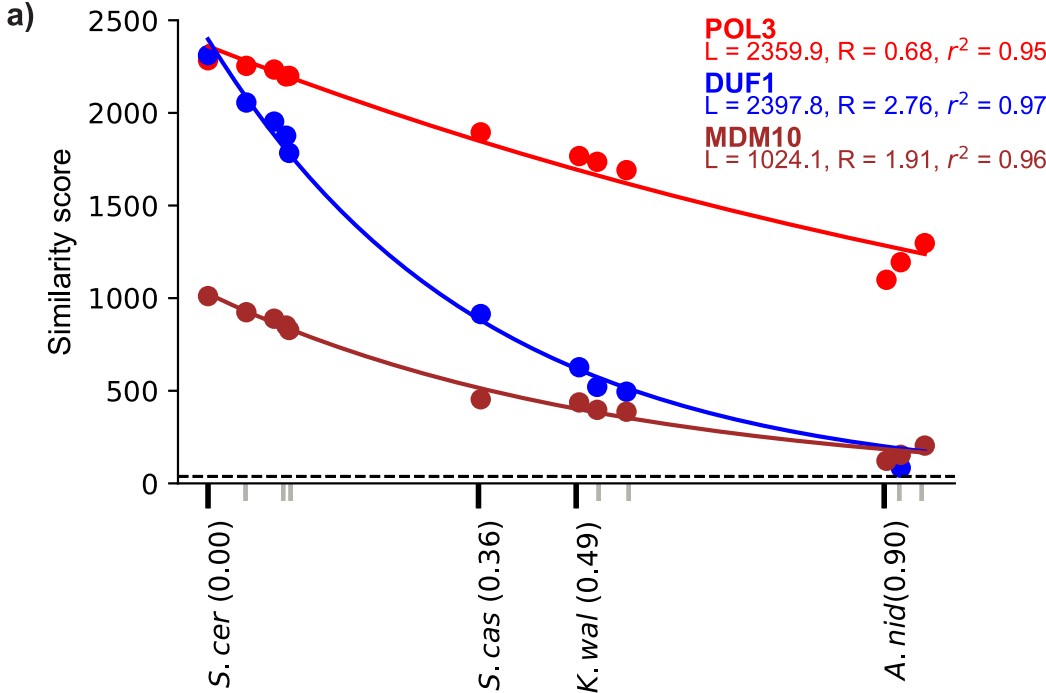

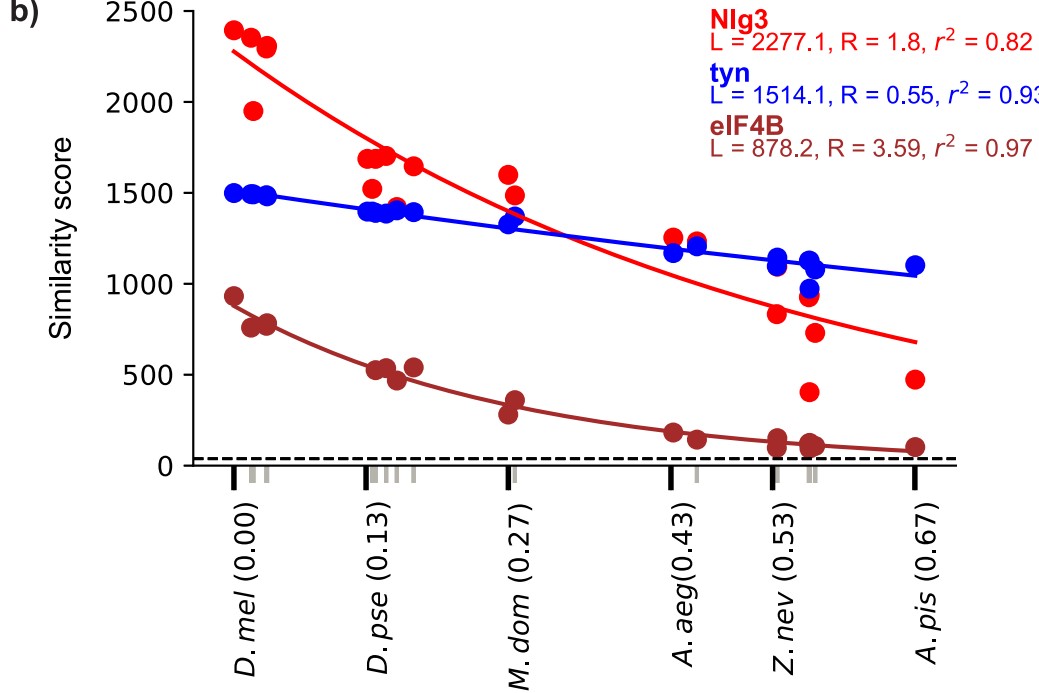

**Fig 1. Depictions of the fit of the null model of similarity score, as defined in the text, decline with evolutionary distance for 3 representative proteins from *Saccharomyces cerevisiae* (a) and *Drosophila melanogaster* (b).** Colored points represent the BLASTP score between the protein and its ortholog in the species that is at the evolutionary distance indicated on the x-axis. Tick marks on the x-axis represent each of the species used here. For visual clarity, only some species names and evolutionary distances are included, indicated with black tick marks; gray tick marks represent the other

unlabeled species. The dashed line represents the detectability threshold, the score below which an ortholog would be undetected at our chosen E-value of 0.001. The best-fit values of $a$ and $b$ are shown for each protein. The $r^2$ value is also shown and was calculated from a linear regression of the log of the similarity score versus evolutionary distance. All data in these figures are available at https://github.com/caraweisman/abSENSE, under Fungi_Data (panel a) and Insect_Data (panel b).

the comparison of the protein with itself.) We identified orthologs using reciprocal best BLASTP search with a threshold of E < 0.001 (Methods). Reciprocal best BLASTP is not a perfect means of distinguishing orthologs from paralogs, and results in some genes failing to be assigned to orthologs in some species, but it suffices for the purpose and is easy to do at scale.

With these similarity scores ($S$) and evolutionary distances ($t$) in hand, we tested how well our model explains the observed decline in similarity scores with increasing evolutionary distance in fungal and insect genes. Our model predicts a linear relationship between the log of ortholog similarity scores and evolutionary distance. We therefore assessed the fit of the model by performing a linear regression of the log of each protein's similarity score, ln $S(t)$, against the inferred evolutionary distance to the focal species, $t$, and computing the square of the Pearson correlation coefficient ($r^2$), which measures how much of the variance in ln $S(t)$ is explained by $t$.

The model predicts similarity scores reasonably well. The mean and median $r^2$ were 0.92 and 0.95 for similarity scores of *S. cerevisiae* genes (S1 Fig). We repeated this with *D. melanogaster* proteins and their orthologs in the other insects, where the mean and median $r^2$ were 0.84

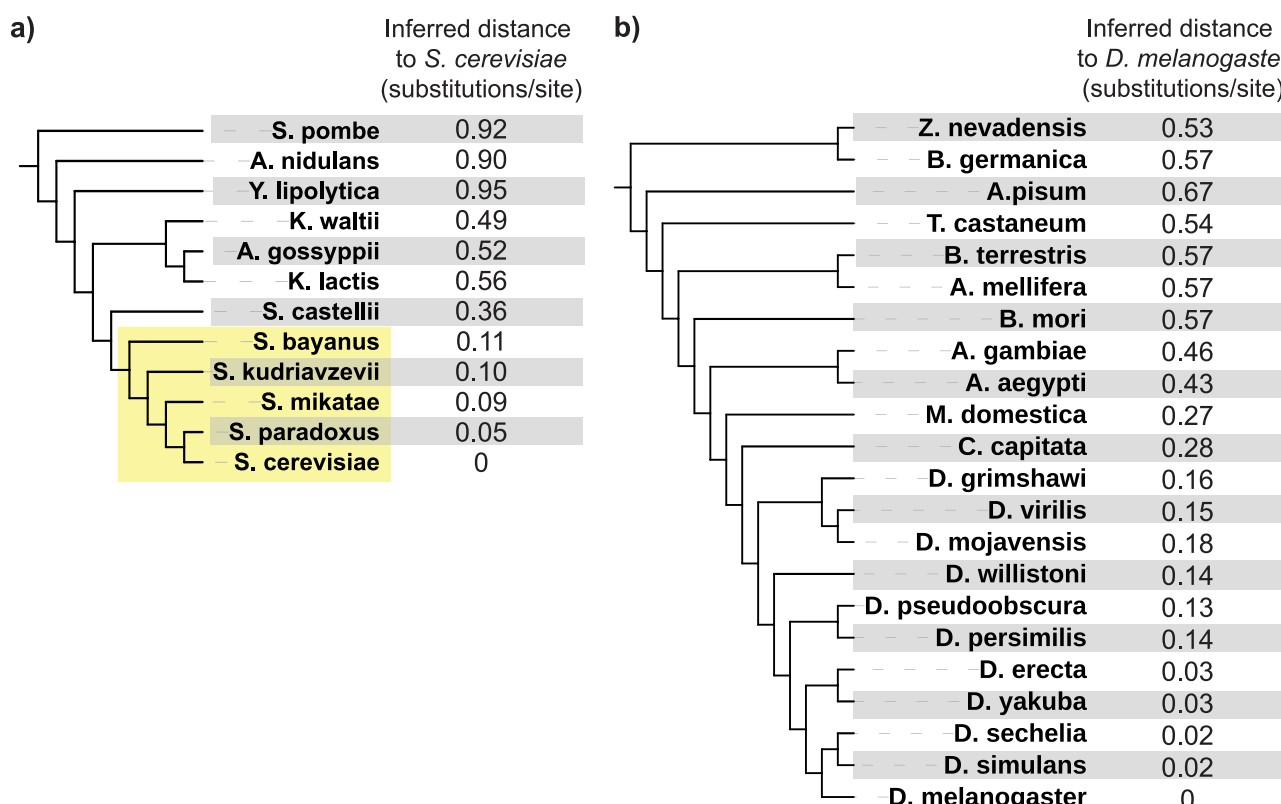

**Fig 2. Inferred evolutionary distances between each fungal species and *S. cerevisiae* (a) and each insect species and *D. melanogaster* (b).** The tree topologies for these taxa are based on previously published studies [34, 35] and were not calculated here; branch lengths are not to scale. The fungal *sensu stricto* lineage, referenced frequently in the text, is shaded in yellow.

and 0.91, respectively, for similarity scores of *D. melanogaster* genes (S1 Fig). Results were similar using the 2 other sets of estimated distances (S1 Fig).

As well as considering the fit of each gene to the expected value of the model, we tested how well our estimate for the variance of the similarity score captured the observed scatter around this expected value. To do this, for the ortholog of each *S. cerevisiae* gene in each species, we calculated the difference between the actual and expected similarity score and expressed it as a multiple of the predicted standard deviation $\sigma = \sqrt{L(1 - e^{-Rt})(e^{-Rt})}$ of the similarity score (a Z-score). We expect these Z-scores to follow a normal distribution if our model's estimated variance is correct, which is roughly what we observe (S2 Fig). Approximately 92% of *S. cerevisiae* orthologs have observed scores within 3 SD of the prediction; for a standard normal distribution, 99% are expected. Of the remaining 8% of scores, 7% are below 3 SD, and 1% are above 3 SD. Results in *D. melanogaster* are similar: 88% have observed scores within 3 SD, 8% below, and 4% above. There is some skew toward predicted scores that are higher than observed scores. We attribute this to the fact that our model neglects how insertions and deletions may disrupt the length of a local alignment. Results were similar when using the 2 other sets of estimated distances (S2 Fig).

Our method strictly requires only 2 orthologs, possibly covering only a very short evolutionary distance, to estimate *L* and *R*. To assess how well these parameters can be estimated in this worst-case scenario, we next asked how similar the values of *L* and *R* inferred from only 2 species very closely related to *S. cerevisiae* (*S. paradoxus* and *S. mikatae*, at evolutionary distances of 0.02 and 0.09 substitutions/site, respectively) were to those inferred above, using all orthologs available in the full set of 11 fungal species, including the most distant species *Schizosaccharomyces pombe* at a distance of 0.92 substitutions/site. We find that concordance between these estimates is good. The $r^2$ and average percent difference between these estimates is 0.99 and 0.6% for the L parameter, respectively, and 0.78 and 8% for the R parameter.

Finally, we asked whether the best-fit values of the parameters *L* and *R* found for the fungal proteins are correlated with the interpretation of these parameters in our model. We expect a protein's value of *L* to be related to its length and *R* to be related to evolutionary rate. For all *S. cerevisiae* genes, we plotted *L* versus protein length and *R* versus evolutionary distance in substitutions/site from multiple alignments of each protein from *S. cerevisiae* and the 4 most closely related species (Methods). The *L* parameter is indeed highly correlated with gene length ($r^2 = 0.99$), and the *R* parameter is more weakly correlated with gene evolutionary rate ($r^2 = 0.51$) (S3 Fig). We attribute some of this lower correlation to the fact that *R*, which describes how quickly score declines, includes the effects of insertions and deletions as well as substitutions, whereas standard measures of evolutionary rate derived from alignments (like the gene evolutionary rates we calculated to compare with *R*) only consider substitutions. The distributions of the estimated *L* and *R* parameters across all genes are long-tailed and approximately log-normal (S4 Fig), consistent with other analyses of distributions of gene length [37] and evolutionary rate [38].

## Many lineage-specific genes can be explained by homology detection failure

Having validated our null model for similarity score decline, we then focused on lineage-specific genes and used the model to ask our central question: How often is homology detection failure alone enough to explain a lineage-specific gene?

We first considered annotated *S. cerevisiae* proteins that are lineage-specific to the *sensu stricto* yeasts, a young lineage sharing a common ancestor approximately 20 million years ago (Mya) containing the 5 species *S. cerevisiae*, *S. paradoxus*, *S. mikatae*, *S. bayanus*, and *S. uvarum* (Fig 2a), which has been the focus of previous work on lineage-specific genes [11, 31].

We identified 375 such *sensu stricto*–specific genes, defined as having homologs detectable by BLASTP in at least one of these species but lacking detectable homologs in the nearest out-group *S. castellii* or in any other outgroups according to a permissive E-value threshold of 0.001 (Methods). Between 40% and 70% of *sensu stricto*–specific genes identified in 2 previous studies are included in this set [11, 31]. The remainder are either open reading frames (ORFs) not used in our initial search because they are marked as dubious in both the Saccharomyces Genome Database and Refseq and so have been removed from the *S. cerevisiae* Refseq annotation, or because we detected homologs outside of the *sensu strictos*, likely due to our permissive E-value threshold. Because our detectability model is regression-based, a minimum of 3 observed homologs (including the gene in the focal species) are required; for example, we could not perform this computation on the *S. cerevisiae* gene *BSC4* [39], proposed to have a very recent de novo origin and thus only found in *S. cerevisiae*. We applied our model to the 155 such *sensu stricto*–specific proteins.

For each of these 155 lineage-specific genes, we used the best-fit values of the *L* and *R* parameters found here previously to extrapolate and predict the score of an ortholog at the evolutionary distance of *S. castellii* under the null model. Using parameters from the *sensu stricto* lineage to extrapolate to more distant species corresponds to assuming that these 2 groups of orthologs have evolved in the same manner since their divergence from their common ancestor. Finally, we calculated the probability that a homolog at the evolutionary distance of *S. castellii* would be detected, P(detected | null model, $t_{castellii}$), by using our model for similarity score variance to generate a probability distribution for the score and computing the percentage of the probability mass in this distribution below our chosen detectability threshold (corresponding to an E-value of 0.001).

This analysis is illustrated for one example of a *sensu stricto*–restricted *S. cerevisiae* protein, Uli1, in Fig 3. Uli1 has been implicated in the unfolded protein response [40], making it one of only a few *sensu stricto*–specific genes with experimental evidence of function, and its lineage specificity has prompted previous studies to propose that it originated de novo [11, 31]. How-ever, we find that the probability that an ortholog of this gene would be detectable in *S. castellii*, P(detected | null model, $t_{castellii}$), is approximately 0, indicating that a null evolutionary model is sufficient to explain the lineage specificity of this short and rapidly evolving gene.

The result of performing this test on all of the 155 *sensu stricto*–specific genes amenable to our analysis is shown in Fig 4a, which depicts the distribution of probabilities of detecting a homolog in the outgroup *S. castellii* given the null model and the evolutionary distance between *S. cerevisiae* and *S. castelli*, P(detected | null model, $t_{castellii}$). Many genes have a very high probability of being undetected, and a majority are more likely to be undetected than detected: 55% have P(detected | null model, $t_{castellii}$) below 0.05, and 73% are below 0.5. This implies that homology detection failure is sufficient to explain a large number, potentially a majority, of these lineage-specific genes. Homologs of these genes only being detected in *sensu stricto* species does not require invoking evolutionary novelty.

We repeated this procedure for *D. melanogaster* genes restricted to the *Drosophila* genus. This young lineage shared a common ancestor approximately 70 Mya, with the housefly *Musca domestica* as the nearest outgroup in our analyses (Fig 5a). We identified 1,611 *Dro-sophila*-restricted genes (Methods), of which 1,278 had the 2 identified orthologs in the *Dro-sophila* lineage required for our analysis. Again, many of these *Drosophila*-restricted genes are very likely to be undetected: 46% have values of P(detected | null model, $t_{domestica}$) below 0.05, and 76% are below 0.5 (Fig 5a). Homology detection failure is therefore also sufficient to explain many lineage-specific genes in this group.

As both the *sensu stricto* yeasts and the drosophilid flies are relatively young lineages, we asked whether these results generalize to older lineages. In fungi, we tested 2 additional

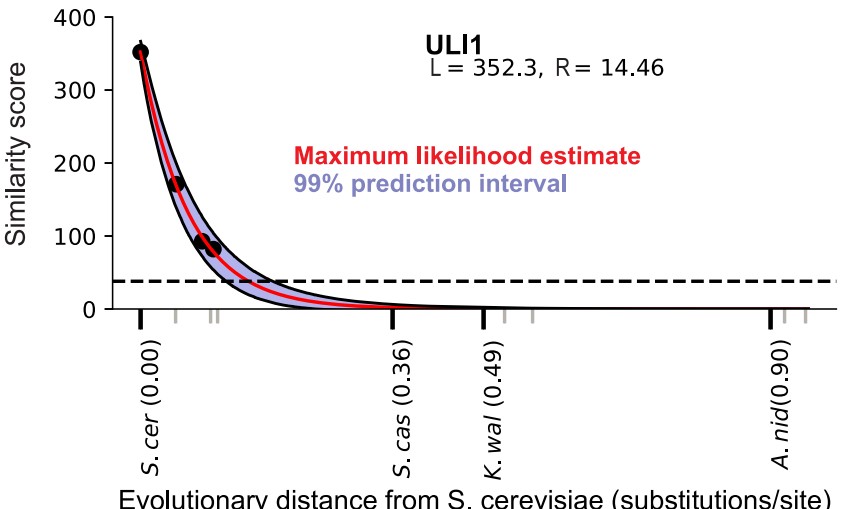

**Fig 3. Illustration of the prediction of detectability decline for the *S. cerevisiae* protein Uli1, displayed as in Fig 1.** At the evolutionary distance of the nearest outgroup *S. castellii*, the entire prediction interval lies below the detectability threshold, indicating an approximately 0% probability that an ortholog would be detected under the null model even if an *S. castellii* ortholog were present. Data in this figure are available at https://github.com/caraweisman/abSENSE/tree/master/Fungi_Data.

lineages with approximate divergence times of approximately 70 Mya (Fig 4b) and approximately 250 Mya [33] (Fig 4c). In insects, we also tested 2 additional lineages, with approximate divergence times of approximately 150 Mya (Fig 5b) and approximately 350 Mya [35] (Fig 5c). We identified all genes specific to each of these 4 additional lineages and then calculated P (detected | null model, $t_{outgroup}$) for all genes with the required 2 identified orthologs, exactly as described for the 2 aforementioned lineages. Results in all of these comparisons are very similar to those in the younger lineages tested here: we predict that a large number of lineage-specific genes have very low probabilities of being detected, with a majority more likely to be undetected than detected (Figs 4b, 4c, 5b and 5c). Homology detection failure is thus sufficient to explain a large number of lineage-specific genes in these older lineages as well. All genes specific to the 6 lineages considered here and their values of P(detected | null model, $t_{outgroup}$) can be found in S3 Table.

As a control, we asked our model to predict the probability of detecting homologs of genes that are not lineage-specific, meaning that these genes have homologs that are detected both inside and outside of the lineage. We repeated the same procedure on all non-lineage-specific genes in the 6 lineages tested here. As we did for the lineage-restricted genes, we used only similarity scores from orthologs within the given lineage to calculate the probability of detecting homologs in the nearest outgroup to the lineage, P(detected | null model, $t_{outgroup}$). If our model operates correctly, it should predict high values of P(detected | null model, $t_{outgroup}$) for these genes, because their homologs are, in fact, detected. In accordance with this expectation, our model predicts that the vast majority (>97% in all lineages) of these genes have a very high probability of being detected, P(detected) > 0.95 (Figs 4 and 5). This analysis, like earlier analyses, was robust to the use of different sets of genes for calculating evolutionary distances (S4 Table).

We separately considered the set of 784 *S. cerevisiae* genes marked as "dubious" in the Saccharomyces Genome Database [41]. Although they have been deemed unlikely to encode functional proteins, many of them are lineage-specific and so have been included in previous

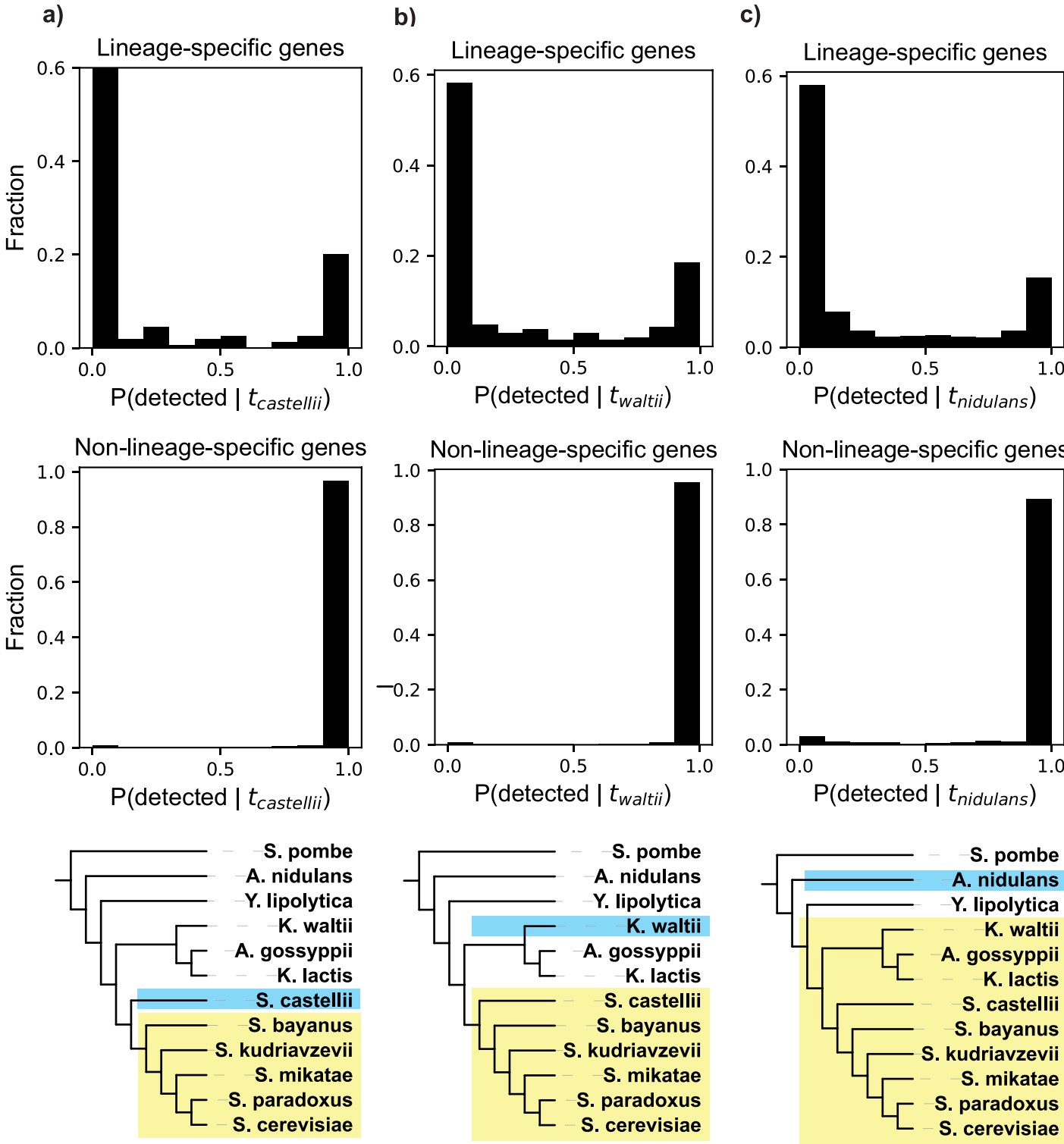

**Fig 4. Distributions of detectability prediction results for 3 yeast lineages (a, b, c).** Top: results for all lineage-specific genes. Middle: results of the same analysis for all non-lineage-specific genes, which serve as a positive control. These genes, which have detectable orthologs outside of the lineage, should be predicted to be detected, which they largely are. Bottom: depiction of the lineage (yellow) and closest outgroup (blue) considered in the analyses in the corresponding column. In c), note that *Yarrowia lipolytica* is the topological outgroup to the shaded lineage but is not the closest species by evolutionary distance (branch lengths are not to scale). Data in this figure are available at https://github.com/caraweisman/abSENSE/tree/master/Fungi_Data.

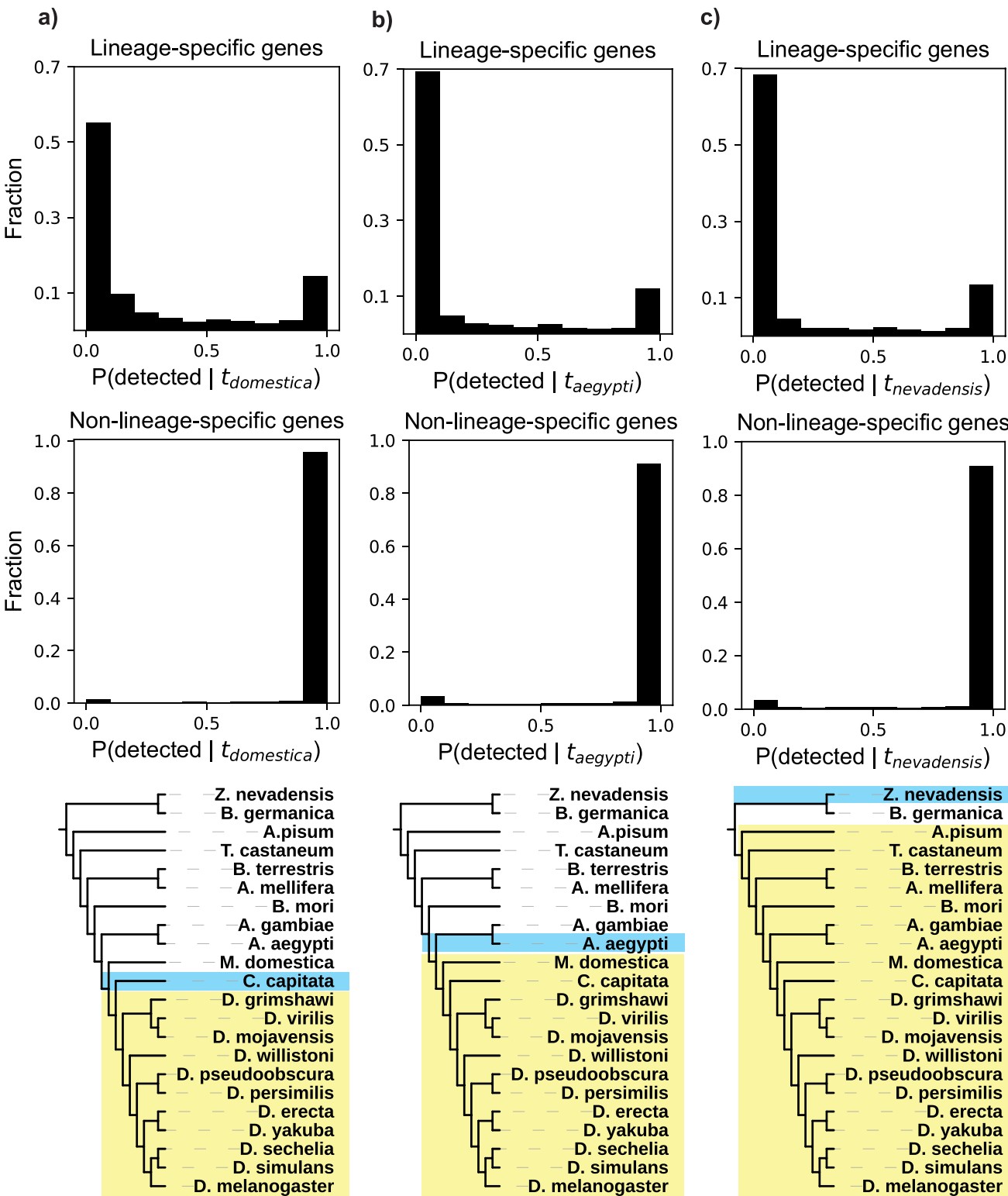

**Fig 5. Distributions of detectability prediction results for 3 insect lineages (a, b, c).** Top: results for all lineage-specific genes. Middle: results of the same analysis for all non-lineage-specific genes, which serve as a positive control. These genes, which have detectable orthologs outside of the lineage, should be predicted to be detected, which they largely are. Bottom: Depiction of the lineage (yellow) and closest outgroup (blue) considered in the analyses in the corresponding column. In a), note that *Ceratitis capitata* is the topological outgroup to the shaded lineage, but is not the closest species by evolutionary distance (branch lengths are not to scale). Data in this figure are available at https://github.com/caraweisman/abSENSE/tree/master/Insect_Data.

studies as potentially novel genes [11]. We analyzed the 167 of these dubious genes that met our analysis requirement of having detected orthologs in at least 2 other species (many are unique to *S. cerevisiae*). We find that homologs of these genes would be undetected at an even higher rate than for validated genes; in all 3 fungal lineages, at least 99% of these dubious ORFs have P(detected | null model, $t_{outgroup}$) below 0.5, and at least 80% are below 0.05 (S5 Table).

Another recent paper used a different approach to estimate the fraction of lineage-specific genes that are attributable to homology detection failure [28]. Vakirlis and colleagues used a small set of genes in "microsyntenic blocks" to count how often a gene is not recognizably similar to its presumptive homolog in the syntenic position in a comparative genome. Assuming that this sample is representative, and so approximates the frequency at which homologous genes in general diverge beyond recognition, they conclude that 20%–45% of lineage-specific genes in fungal, insect, and vertebrate phylogenies are attributable to homology detection failure. We consider this result to be in qualitative agreement with ours, concluding that homology detection failure generates a substantial fraction of lineage-specific genes. However, we produce somewhat larger estimates of the rate of detection failure. We investigated the cause of this discrepancy in additional analyses described in detail in S1 Supplemental Info. Briefly, we find that genes in microsyntenic regions evolve more slowly than those outside of such regions, so using this subset of genes leads to a lower inferred rate of detection failure. Consistent with this observation, when we restrict our analysis in fungi to consider only those lineage-specific genes that are within microsyntenic regions, we find a lower rate of homology detection failure that is approximately consistent with the estimate of Vakirlis and colleagues.

## More sensitive homology searches detect beyond-lineage homologs for many lineage-specific genes well-explained by homology detection failure

If a gene being lineage-specific is due to the failure of BLASTP to detect homologs that are in fact present, we would expect that a more sensitive search will sometimes succeed in finding homologs where BLASTP did not. We asked whether this was the case for genes whose lineage specificity was consistent with the hypothesis of detection failure: Can we use a more sensitive method to find previously undetected homologs for these genes? We refer to such homologs, detected using a different method in species outside of the originally defined lineage, as "beyond-lineage homologs."

We used *sensu stricto* yeast–specific genes as a case study to ask this question. These yeasts and several of their nearest outgroups have a high degree of conservation of chromosomal gene order (synteny), presenting the opportunity for a more sensitive search. A standard similarity search tests all proteins in a large database of sequences, such as a complete proteome. The resulting multiple testing burden requires a higher score to achieve statistical significance than would be required for a search over a smaller number of sequences. In these yeasts, synteny allows us to restrict a similarity search to 1 candidate gene at the orthologous chromosomal locus, reducing the multiple testing burden and enabling ortholog identification with a lower score. For the fungal species used here, a proteome-wide search would need a BLASTP score of approximately 37 to achieve an E-value of 0.001, but a single-protein search would only require a score of approximately 24. Orthologs with scores between these 2 values would be missed in our initial search but successfully detected with synteny-guided similarity searches.

We used this strategy to search for beyond-lineage orthologs for all *sensu stricto*–specific genes for which the null model of detection failure is a reasonable explanation. We use a threshold of P(detected | null model) < 0.95 to define these genes. This choice is a conservative threshold that corresponds to genes that are insignificant according to a traditional

significance test threshold of P(undetected | null model) = 1 −P(detected | null model) > 0.05. There are 126 *sensu stricto*–specific genes that pass this threshold.

To identify the orthologous locus in outgroup yeasts for these 126 *S. cerevisiae* genes, we used the Yeast Gene Order Browser (YGOB), an online resource that curates the chromosomal orthology relationships between species including the *sensu stricto* yeasts, *S. castellii*, *Kluyveromyces waltii*, *Ashbya gossypii*, and *K. lactis* [42]. Of these 126 *sensu stricto*–specific genes, 19 are included in YGOB and have an orthologous locus in at least one of these outgroup yeasts. For all of these genes, the upper bound of the 99% prediction interval for the similarity score predicted by our model is above the detectability threshold of 24 bits, indicating that they are potentially detectable by this analysis. Of these 19 genes, 17 had an annotated gene at the orthologous locus in at least one outgroup species. For 11 of these, at least one of these genes at an outgroup orthologous locus had significant detectable similarity (E < 0.001) to the *S. cerevisiae* gene. In all but 2 of these cases, the similarity score fell within our prediction interval (in those 2 cases, the similarity score was slightly higher than predicted). These 11 genes and their proposed orthologs are listed in S5 Table.

In total, we found beyond-lineage homologs for 46% of genes for which we were able to perform a synteny analysis. We note that this is a conservative estimate. We only considered ORFs that are already annotated in outgroup species, although unannotated orthologs may be present. Additionally, the lower bound of the 99% similarity score prediction interval for all remaining 54% of these genes is lower than the threshold required for detection via synteny, so that all have some probability of orthologs still being missed in this analysis.

## Some lineage-specific genes are poorly explained by homology detection failure

In all lineages studied here, there are also lineage-specific genes that are poorly explained by the null hypothesis: Their similarity scores decline too slowly to make homology detection failure alone a good explanation for their lineage specificity. These are the genes with high values of P(detected | null model). In all 6 lineages we studied, 10%–20% of lineage-specific genes have detection probabilities of 0.95 or greater (Figs 4 and 5).

This result is illustrated by one such *sensu stricto*–specific protein, Spo13, in Fig 6. Spo13 has been proposed as a candidate de novo gene [31] by virtue of its lineage specificity, and this analysis highlights it as a particularly promising novel gene candidate among the large number of other lineage-specific genes in the *sensu stricto* lineage.

The existence of lineage-specific genes like Spo13, which our null model predicts should have detectable homologs outside of the lineage, indicates that evolutionary mechanisms beyond those included in the null model may be operating. Among such mechanisms are those postulated by the novelty hypothesis, like de novo origination and duplication-induced neofunctionalization. However, other known mechanisms could also explain such genes. These include processes that cause the gene tree to deviate from the species tree, like horizontal gene transfer and any mechanisms that change the evolutionary rate of a protein on a restricted part of the tree.

## Characterization of yeast lineage-specific genes that are poorly explained by homology detection failure

We next aimed to characterize genes whose lineage specificity is poorly explained by homology detection failure. We again used *sensu stricto*–specific genes as a case study, allowing for synteny analysis and the biological insight provided by many genes in *S. cerevisiae* being comparatively well-studied. We selected the subset of *sensu stricto*–specific genes, including Spo13,

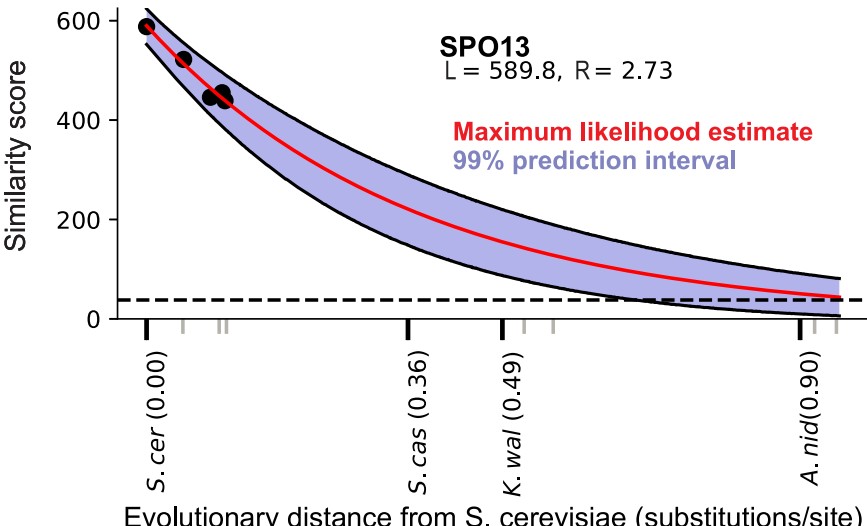

**Fig 6. Detectability prediction results for the *S. cerevisiae* protein Spo13, displayed as described in Fig 3.** At the evolutionary distance of the nearest outgroup *S. castellii*, the entire prediction interval lies well above the detectability threshold, indicating an approximately 100% probability that an ortholog should be detected in this species under the null model. Data in this figure are available at https://github.com/caraweisman/abSENSE/tree/master/Fungi_Data.

whose lineage-specificity is poorly explained by homology detection failure, i.e. for which P (detected | null model) > 0.95. These are genes on the other side of the threshold applied above: the null hypothesis strongly predicts that homologs should be detected, making their lineage specificity incompatible with the null hypothesis. There are 25 *sensu stricto*–specific genes that satisfy this threshold. Although a thorough study of these genes is beyond our scope, we report a few initial observations.

"De novo origination," the process of a new gene emerging from previously noncoding sequence, is a commonly proposed origin of lineage-specific genes [12]. We asked how many of these 25 lineage-specific genes could plausibly be such de novo genes. By definition, genes that have emerged de novo in the *sensu stricto* lineage should have no out-of-lineage homologs, and so a more sensitive synteny-based homology search strategy should fail to find such homologs. We performed a synteny-based search for out-of-lineage homologs for these 25 genes in the same way as described for genes that are well-explained by detection failure. For 20 of these 25 genes, an orthologous locus is listed in YGOB. Of these 20, 12 have annotated genes with significant similarity (E < 0.001) at the orthologous locus in at least 1 outgroup species. Thus, 12 of 25 genes, or just under half, of genes that are not well explained by homology detection failure did not originate de novo in the *sensu stricto* lineage. This is a conservative estimate of the total number of genes that have out-of-lineage homologs, because, as described here, even this synteny-based homology search has finite sensitivity. Spo13, the gene shown in Fig 5, is one example of these lineage-specific genes that nonetheless are not de novo originated: It has out-of-lineage orthologs identifiable by synteny in *S. castellii*, *K. waltii*, *K. lactis*, and *A. gossypii*.

Genes that acquire a new function following duplication and divergence ("neofunctionalization") are another proposed source of lineage-specific genes [12]. We therefore asked how many of our *sensu stricto*–specific genes have a paralog, consistent with the hypothesis that they emerged through duplication and divergence. Based on BLASTP searches within the *S. cerevisiae* genome, we find that 4 of the 25 lineage-specific genes have annotated paralogs specific to some subset of the *sensu stricto* yeasts, which therefore likely emerged after their

divergence from *S. castellii*. We also find using YGOB that another 4 of these 25 genes have annotated paralogs resulting from the yeast whole-genome duplication, which occurred before the divergence of *S. castellii* from the *sensu stricto* yeasts. In total, 8/25, or fewer than one third, of these genes show evidence of having been the result of duplication events. However, we note that this estimate for the number of genes with paralogs is again conservative because of the finite sensitivity of the homology searches.

Finally, we performed a gene ontology enrichment test (Methods) to determine if certain biological processes were statistically overrepresented among these 25 genes. We find significant enrichment of genes involved in several GO categories relating to spore formation and meiosis, including "ascospore-type prospore membrane assembly" ($p = 7^*10^{-5}$; 3 observed versus 0.7 expected) and "meiotic cell cycle process" ($p = 5^*10^{-5}$; 7 observed versus 1 expected). Spo13, involved in meiotic cell cycle regulation through its roles in maintaining sister chromatid cohesion during meiosis I and promoting kinetochore attachment [43], is one such example. By contrast, no biological processes were overrepresented among lineage-specific genes that are consistent with homology detection failure (although these genes are much less likely to have GO annotations at all: 92% have no annotation, compared with 12% of all *cerevisiae* genes and 36% of lineage-specific genes that are inconsistent with homology detection failure).

A table of these 25 genes and the features discussed in this section can be found in S6 Table.

## Discussion

The widespread interpretation of lineage-specific genes as evolutionarily novel assumes that absence of evidence for detectable homologs in outgroups is evidence that homologs are absent. The model we have presented here allows us to formally test the alternative, null hypothesis: Homologs do exist outside the specified lineage, but they have diverged, at a constant novelty-free evolutionary rate, beyond the ability of a similarity search program to detect them. We find that this hypothesis is sufficient to explain a large number of lineage-specific genes in 2 taxa in which lineage-specific genes have been interpreted as exhibiting some kind of evolutionary novelty. These results caution against automatically assuming that lineage-specific genes are novel.

Two important caveats should be kept in mind. First, this method cannot exclude the possibility that a gene is truly novel, but also short enough and evolving fast enough that its ortholog would not be detected if present, such that homology detection failure can also explain its lineage specificity. For this reason, it may be difficult for de novo genes in particular, which have been hypothesized to be short and fast-evolving, to reject the null model. However, in this case, where 2 hypotheses can both explain a lineage-specific gene, we argue that additional evidence should be required to prefer the comparatively exotic hypothesis of novelty to the more conservative one of detection failure. Our case study in the *sensu stricto* yeasts finds that more sensitive synteny-based homology searches successfully find previously undetected homologs for many lineage-specific genes, supporting this argument. Second, these results may or may not generalize to classes of lineage-specific genes that we have not considered here. Because our method requires at least 2 observed orthologs, we have only applied it here to genes found in at least 2 species in the lineages in question. Moreover, like other studies, we focused on genes in existing annotations, which are prone to biases that may exclude novel genes. However, when we analyze *cerevisiae* ORFs with the requisite 2 orthologs that are marked as of "dubious" coding status in the Saccharomyces Genome Database, we find that an even larger proportion—nearly all—are unlikely to be detected. Although we have not done so here, we note that this method could be extended to these classes of genes. In principle, individual conspecifics with sufficient genetic differentiation could be used as discrete taxa in our method to

analyze genes found in only 1 or 2 species. Additionally, the method is readily applicable to any protein annotation, which could be custom-made as desired.

Although we find that many lineage-specific genes can be adequately explained by homology detection failure, we also find a minority of lineage-specific genes in fungi and insects that cannot. This leaves open the possibility that these genes are biologically novel. However, the reason that these genes reject our null model is not addressed by our present work. Our initial analyses do show that many of these genes are neither de novo genes nor have detectable paralogs, suggesting that processes other than the commonly proposed hypotheses of de novo origination and duplication-divergence may be at play. There are many possible processes that could cause genes to deviate from our null model, but one speculative example lies in the observed enrichment in yeast of genes involved in meiotic processes, exemplified by Spo13. This strikes us as suggestive of meiotic drive phenomena, which have been observed in yeast [44] and have been shown to cause rapid protein divergence [45], producing clade-specific rate accelerations leading to lineage-specific genes. More detailed characterization of these genes is required to understand if and in what way they are evolutionarily novel.

There is increasing consensus that homology detection failure is frequent [28]. It should be taken into account in studies that aim to use lineage-specific genes to identify candidates for genetic novelty. To this end, our approach allows us to determine whether a particular lineage-specific gene is attributable to homology detection failure. Synteny analyses of the kind used here can sometimes be used to determine whether out-of-lineage orthologs are present [46, 47] and can provide strong evidence of de novo origination [48], but syntenic analyses are only possible in the limited taxa where sequenced species are related closely enough that synteny is conserved. By contrast, our method can be used in any set of species for which relative evolutionary distances are known. We expect it to be useful in the wide variety of studies that aim to identify novel genes that may underlie the evolution of morphological, behavioral, and other novel traits [7, 49–53]. An implementation of our method, with all raw data and results presented here, is freely available as source code at github.com/caraweisman/abSENSE and as a web server at eddylab.org/abSENSE.

## Methods

### Identification of *S. cerevisiae* and *D. melanogaster* orthologs

We downloaded previously annotated proteomes of all species used here from several sources, largely Refseq and GenBank. Accession IDs for Refseq and GenBank proteomes and download links for those from other sources are listed in S2 Table. We performed a BLASTP (version 2.8.0) search [54] with an E-value threshold of 0.001 using the *S. cerevisiae* proteome as the query against each of the 11 other yeast proteomes independently. We also performed the reciprocal of each of these searches, using each of the 11 other yeast proteomes as the query against the *S. cerevisiae* proteome. We used a custom Python script to identify reciprocal best BLAST hits for each *S. cerevisiae* protein in each of the other yeast proteomes. A protein in another yeast's proteome was considered a reciprocal best hit to the *S. cerevisiae* protein if (1) the E-value of the *S. cerevisiae* protein against that protein was the lowest of any in that species' proteome and (2) the E-value of that protein against the *S. cerevisiae* protein was the lowest of any protein in the *S. cerevisiae* proteome. Proteins in the other yeast species satisfying this reciprocal best hit criterion were considered orthologs of the *S. cerevisiae* protein. When no significant homology to a *S. cerevisiae* protein was detected in another species, or when the reciprocal best hit criterion was not met by any protein in that species, no ortholog was assigned in that species. To identify orthologs for *D. melanogaster* proteins, we repeated this

same procedure for all *D. melanogaster* proteins and each of the 21 other insect species' proteomes.

## Calculation of evolutionary distances

Because evolutionary distance *t* only appears in our model as a product with the gene-specific rate parameter *R*, we can use a subset of genes in the species group to infer these relative distances. Each gene's value of *R* will scale these relative distances appropriately when fit to the model: Genes that evolve faster than these relative distances will have values of *R* above 1, and slower, below. We used BUSCO genes as the subset of genes from which to estimate distances, as they are generally well-conserved, facilitating ortholog identification and alignment. This enables our desired result of a species tree with correct relative evolutionary distances (in substitutions/site across aligned BUSCO genes), which is the only feature needed by our downstream inference. We downloaded a list of eukaryotic BUSCO genes [36] from the BUSCO web server (https://busco.ezlab.org/) and identified all of these genes for which we were able to identify an ortholog of the corresponding *S. cerevisiae* gene in all 11 other yeast species ("Identification of orthologs"). We found 102 such BUSCO genes. We used the alignment software MUSCLE (version 3.8.31) [55] with default parameters to create a multiple sequence alignment of the orthologs from all 12 yeast species for of each of these 102 genes. We then concatenated these alignments and used the Protdist program from the PHYLIP software package (version 3.696) [56] with default parameters to find pairwise evolutionary distances for all 12 yeast species in substitutions per site. To test the effect of using a smaller number of genes to infer these distances, we then randomly and independently selected 2 subsets of 15 of these 102 genes and performed the same alignment and distance calculation procedure on each of these 2 subsets. We then performed the same procedure using *D. melanogaster* genes and the 21 other insect species. Here, there were 125 BUSCOs for which we were able to identify orthologs in all species, and the 2 random subsets of 15 genes were selected from among these 125. Refseq accessions for genes in the 3 sets of BUSCOs in both taxa are listed in S7 Table.

## Correlation of *R parameter* with evolutionary rate

To determine the correlation between each gene's best-fit value of the *R* parameter in our model and the substitution rate, we used alignments of 5,261 *S. cerevisiae* genes and their orthologs in all 4 other *sensu stricto* yeast species generated by a previous study [57]. We opted not to include more distantly related species in these alignments for the sake of more reliable ortholog identification and alignment construction. We used the protdist function of the PHYLIP package (version 3.696) [56] on these alignments to infer the number of substitutions per site between the *S. cerevisiae* gene and its ortholog in the most distant *sensu stricto* yeast *S. kudriavzevii* (we chose a fairly distant representative of these species to minimize sampling error from low substitution counts) and correlated this value with the *R* parameter inferred from the regression analysis.

## Identification of lineage-specific genes

To identify *S. cerevisiae* genes specific to the 3 yeast lineages tested here, we performed a BLASTP search [54] with an E-value threshold of 0.001 for each gene in the *S. cerevisiae* proteome as the query against each of the 11 other yeast proteomes independently, using the same proteomes listed in S2 Table. If the BLASTP search detected no homologs of the *S. cerevisiae* gene in the proteomes of any of these species outside of the specified lineage, we considered it

lineage-specific. We applied the same criterion using the 21 other insect proteomes to identify *D. melanogaster* genes specific to the 3 insect lineages tested here.

## Synteny-based homology searches

We used version 7 of the YGOB's online web tool (http://ygob.ucd.ie/) [42]. For tested *S. cerevisiae* genes, if the gene was included in this YGOB version, we determined whether an orthologous chromosomal region in any of the outgroup yeast species used here had been identified in the browser. If so, we searched for any genes in these outgroup species at the locus that were annotated in the browser. We considered genes to be within the outgroup orthologous locus if they were between the outgroup's orthologs of the closest *S. cerevisiae* genes up- and downstream of the query gene. If annotated genes existed at the orthologous locus, we performed a BLASTP search of the *S. cerevisiae* sequence against the sequences of all outgroup genes at that locus as listed in YGOB and called orthology in cases where this single-search E-value was <0.001.

## Gene ontology analysis

We used the Gene Ontology Consortium's online web server (http://geneontology.org/) [58] to test whether or not certain biological functions were enriched in the set of *sensu stricto*–specific genes that we found to be poorly explained by detection failure. We performed a Fisher's exact test using the "GO biological process complete" annotation data set for all *S. cerevisiae* genes.

## Calculation of substitutions/site and gaps for *sensu strictu* alignments

Although Vakirlis and colleagues (2020) [28] provide dN values computed from alignments of *S. cerevisiae* genes and their *sensu strictu* orthologs, these underlying alignments are restricted to the 5,261 genes for which orthologs were identified in all 5 *sensu strictu* species in a previous study [57]. Because we worried that this might introduce a bias against quickly evolving genes, for which orthologs are less readily identifiable, we opted to make our own alignments, including all 5,586 genes for which we could identify an ortholog in at least one of the 4 other *sensu strictu* species. We used both MUSCLE [55] and ClustalOmega [59] with default settings to produce a multiple alignment of each *cerevisiae* gene and its orthologs in at least one other *sensu strictu* species. We then used these alignments to compute substitutions/site between *S. cerevisiae* and *S. bayanus* with the PHYLIP ProtDist program as in our other evolutionary distance calculations ("Calculation of evolutionary distances" section). We chose *S. bayanus* because it is the most distant species from *cerevisiae* according to our analysis (Fig 2). We then used all genes with an ortholog present in S. bayanus, regardless of its status in the 3 other yeast species, in the subsequent analysis. Results from the 2 alignment programs were extremely similar, as were results using distances to the slightly closer *S. kudriavzevii*.

  For the analysis shown in S1 Supporting Information, we then used these same alignments to count the total number of gaps in each alignment and divide by the number of columns and number of sequences in the alignment to calculate the gaps per column per sequence.

## Detectability prediction analysis of microsyntenic lineage-specific genes

S1 Supporting Information, we aimed to re-perform our original analysis of fungal lineage-specific genes but restricted to genes determined to be in microsyntenic regions by Vakiriis and colleagues (2020) [28]. We included genes in this analysis as follows. We started with the same list of genes specific to the lineages for which *S. castellii* and *K. waltii* are the closest

outgroups (Fig 4) as in our original analysis. From these, we selected genes that Vakirlis and colleagues determined to be in a microsyntenic region in at least one of (1) a species within that lineage; (2) that species itself (*S. castellii* or *K. waltii*); or (3) another outgroup species to the lineage of very similar divergence time to (2). We chose to include genes in microsyntenic regions in species within the lineage and not just in its closest outgroup to be maximally conservative and because the number of genes in microsyntenic regions only in the outgroup species was low. We chose to allow for another outgroup species of similar divergence time to be substituted for the species that we used as outgroup because the set of species for which Vakirlis and colleagues performed a synteny analysis did not overlap exactly with the set of species used here (for example, *K. waltii* was not included), such that this was the closest approximation possible using those data. In the case of *S. castellii*, these species included *S. arboricola*, *S. kudriavzeviii*, and *S. castellii* itself. In the case of *K. waltii*, these species included *K. lactis*, *A. gossypii*, *L. thermotolerans*, *E. cymbalariae*, *A. aceri*, *S. arboricola*, *S. kudriavzeviii*, and *S. castellii*.

## Supporting information

**S1 Table. Inferred distances in substitutions/site from *S. cerevisiae* to each yeast species (top) and from *D. melanogaster* to each insect species (bottom).** Distances were inferred from all BUSCOs with orthologs identifiable in each species group, as well as from 15 genes randomly selected from these BUSCOs. The "15 BUSCOs subset 1" distances were used for all main figures in the text. BUSCO, Benchmarking Universal Single Copy Ortholog. (XLSX)

**S2 Table. Sources of species protein annotations used in this study.** (XLSX)

**S3 Table. All lineage-specific genes and their values of P(detected | null model, $t_{outgroup}$) for the 6 lineages (3 fungi, 3 insect) considered here.** (XLSX)

**S4 Table. Correlation coefficients for gene detectability prediction results based on evolutionary distance estimates derived from 3 different sets of genes (the same as those shown in S1 Fig).** (XLSX)

**S5 Table. List of 11 *S. cerevisiae* genes for which synteny-based searches in YGOB revealed candidate out-of-lineage orthologs, the YGOB IDs of those orthologs, and their synteny search E-values.** YGOB, Yeast Gene Order Browser. (XLSX)

**S6 Table. List of *sensu stricto*–specific *S. cerevisiae* genes that are poorly explained by the hypothesis of detection failure and their features as described in summary in the text.** (XLSX)

**S7 Table. List of RefSeq accession IDs for BUSCOs used in evolutionary distance calculations.** BUSCO, Benchmarking Universal Single Copy Ortholog. (XLSX)

**S1 Fig. $r^2$ distributions for the fit to the model of *S. cerevisiae* and *D. melanogaster* genes using evolutionary distances derived from 3 sets of genes.** a: *S. cerevisiae* genes with distances derived from 102 BUSCOs. **b**: *S. cerevisiae* genes with distances derived from a randomly selected subset of 15 of the BUSCOs used in a. **c**: *S. cerevisiae* genes with distances

derived from a second randomly selected subset of 15 of the BUSCOs used in a. **d**: *D. melanogaster* genes with distances derived from 125 BUSCOs. **e**: *D. melanogaster* genes with distances derived from a randomly selected subset of 15 of the BUSCOs used in d. **f**: *D. melanogaster* genes with distances derived from a second randomly selected subset of 15 of the BUSCOs used in d. In d-f, the peak near $r^2 = 0$ is comprised of genes with orthologs identifiable only in a subset of the closely related *Drosophilid* flies, such that their sequences are identical or nearly identical in all species, except 1 or 2 in which a large chunk of the *melanogaster* protein is absent from the annotation, resulting in almost none of the variance in score (of which there is none, save this large event) being explained by divergence time. We consider this an artifact of the method, as it only appears in the limited cases where the sequences in question are almost totally identical. Data used to generate these figures are available at https://github.com/caraweisman/abSENSE/tree/master/Data_for_supplemental_figures. BUSCO, Benchmarking Universal Single Copy Ortholog. BUSCO, Benchmarking Universal Single Copy Ortholog. (EPS)

**S2 Fig. Distribution of position of BLASTP scores between *S. cerevisiae* and outgroup yeast (top) and *D. melanogaster* and outgroup insects (bottom) relative to the predicted confidence interval.** 0 indicates that the score has the same value as the best fit to the model; multiples of sigma indicate that the score is that many standard deviations above or below the best-fit value. **a**: *S. cerevisiae* genes with distances derived from 102 BUSCOs. **b**: *S. cerevisiae* genes with distances derived from a randomly selected subset of 15 of the BUSCOs used in a. **c**: *S. cerevisiae* genes with distances derived from a second randomly selected subset of 15 of the BUSCOs used in a. **d**: *D. melanogaster* genes with distances derived from 125 BUSCOs. **e**: *D. melanogaster* genes with distances derived from a randomly selected subset of 15 of the BUSCOs used in d. **f**: *D. melanogaster* genes with distances derived from a second randomly selected subset of 15 of the BUSCOs used in d. Data used to generate these figures are available at https://github.com/caraweisman/abSENSE/tree/master/Data_for_supplemental_figures. BUSCO, Benchmarking Universal Single Copy Ortholog. (EPS)

**S3 Fig. Correlation between best-fit parameters and gene properties in yeast.** a: Correlation between each *S. cerevisiae* protein's best-fit value of *a* and its length in amino acids. The *a* parameter is consistently larger than the length due to most identical alignment positions contributing a score larger than 1 according to the scoring scheme used here (BLOSUM62). b: Correlation between each *S. cerevisiae* protein's best-fit value of *b* and its relative evolutionary rate in substitutions per site from *sensu stricto* protein alignments (Methods). Data used to generate these figures are available at https://github.com/caraweisman/abSENSE/tree/master/Data_for_supplemental_figures. (EPS)

**S4 Fig. Distribution of best-fit parameter values for all *S. cerevisiae* proteins.** a: Distribution of the best-fit *a* values for all *S. cerevisiae* proteins. b: Distribution of the best-fit *b* values for all *S. cerevisiae* proteins. Data used to generate these figures are available at https://github.com/caraweisman/abSENSE/tree/master/Data_for_supplemental_figures. (EPS)

**S5 Fig. Results of "dubious" ORF analysis in *S. cerevisiae*.** Top: Distributions of detectability prediction results for all S. cerevisiae lineage-specific genes annotated as of "dubious" coding status in the Saccharomyces Genome Database [41] in 3 yeast lineages (a, b, c). Bottom: Depiction of the lineage (yellow) and closest outgroup (blue) considered in the analyses in the corresponding column. In c), note that *Y. lipolytica* is the topological outgroup to the shaded

lineage, but is not the closest species by evolutionary distance (branch lengths are not to scale). Data used to generate these figures are available at https://github.com/caraweisman/abSENSE/tree/master/Data_for_supplemental_figures. ORF, open reading frame.
(EPS)

**S1 Supporting Information. Supplemental information.** Justification for the functional form of the model; effect of site-specific selection pressure; analysis of data in Vakirlis and colleagues (2020).
(DOCX)

## Author Contributions

**Conceptualization:** Caroline M. Weisman, Andrew W. Murray, Sean R. Eddy.

**Formal analysis:** Caroline M. Weisman.

**Funding acquisition:** Sean R. Eddy.

**Investigation:** Caroline M. Weisman.

**Methodology:** Caroline M. Weisman.

**Resources:** Andrew W. Murray, Sean R. Eddy.

**Software:** Caroline M. Weisman.

**Supervision:** Andrew W. Murray, Sean R. Eddy.

**Validation:** Caroline M. Weisman.

**Writing – original draft:** Caroline M. Weisman, Andrew W. Murray, Sean R. Eddy.

**Writing – review & editing:** Caroline M. Weisman, Andrew W. Murray, Sean R. Eddy.

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
