## [Editor Report · Decision Letter 0]

2 Mar 2020

Dear Dr Eddy, 

Thank you for submitting your manuscript entitled "Many but not all lineage-specific genes can be explained by homology detection failure" for consideration as a Research Article by PLOS Biology.

Your manuscript has now been evaluated by the PLOS Biology editorial staff, as well as by an academic editor with relevant expertise, and I'm writing to let you know that we would like to send your submission out for external peer review. Many thanks for being upfront about the recent eLife paper from the McLysaght group. As you may know, PLOS Biology has a strong "anti-scooping" policy (https://journals.plos.org/plosbiology/article?id=10.1371/journal.pbio.2005203), which means that this will not impact consideration of your study. Indeed, the concordant findings from orthogonal methods are, if anything, a positive.

Please re-submit your manuscript within two working days, i.e. by Mar 04 2020 11:59PM.

Kind regards,

Roli Roberts

Senior Editor

PLOS Biology

---

## [Decision Letter · Decision Letter 1]

6 Apr 2020

Dear Dr Eddy,

Thank you very much for submitting your manuscript "Many but not all lineage-specific genes can be explained by homology detection failure" for consideration as a Research Article at PLOS Biology. Your manuscript has been evaluated by the PLOS Biology editors, an Academic Editor with relevant expertise, and by five independent reviewers. I'm very sorry about this unusually high number of reviewers (we usually aim for 4 or 5), but we wanted to secure some key expertise, and several delivered in very quick succession.

The Academic Editor has asked me to make it clear that given the number of reviewers and the very unusual circumstances surrounding the Covid-19 crisis, we are open to discussion about which of the reviewers' points are essential to address for further consideration, so do feel free to run a revision plan past us.

You'll also see that several of the reviewers wonder whether you should focus your manuscript more around the method. If you choose to go down that route, you might want to consider changing the article type to "Methods and Resources," which has a lesser requirement for novel biological insight. We leave it up to you to decide whether to do this or to keep it as a regular Research Article.

In light of the reviews (below), we will not be able to accept the current version of the manuscript, but we would welcome re-submission of a much-revised version that takes into account the reviewers' comments. We cannot make any decision about publication until we have seen the revised manuscript and your response to the reviewers' comments. Your revised manuscript is also likely to be sent for further evaluation by the reviewers.

We expect to receive your revised manuscript within 2 months. 

**IMPORTANT - SUBMITTING YOUR REVISION**

*Re-submission Checklist*

*Published Peer Review*

*PLOS Data Policy*

*Blot and Gel Data Policy*

Sincerely,

Roli Roberts

Senior Editor

PLOS Biology

REVIEWERS' COMMENTS:

Reviewer #1:

Caroline Weisman and colleagues present a superbly written and reasoned analysis that helps answer an important and timely question in evolutionary genetics: how often do genes evolve from non-protein coding sequence? This question has recently received much theoretical, computational and experimental attention in similarly high-profile journals (e.g., Nature Communications, eLife), so it is an entirely appropriate one for PLOS Biology and should interest many readers. This manuscript represents an important contribution to the field and should result in future researchers exercising considerably increased caution before they claim that a lineage-specific gene has evolved de novo or through an otherwise "exotic" evolutionary process.

Typically, lineage-specific genes (which can include de novo genes, but also other classes) are identified by a lack of BLASTP hits in outgroup species. While such evidence is consistent with genetic novelty, novelty is not the only hypothesis that can explain lineage restriction. Here, the authors present a confident, yet careful, analysis of an important alternative: genes that appear to be lineage specific may actually have homologs in outgroup species that are simply too divergent to detect by algorithms such as BLASTP. Elegant in its simplicity (and explained well enough for even non-computational biologists to understand), the method essentially calculates the rate of sequence similarity decay among the species in which homologs can be found. Then, based on this rate, the method determines in which, if any, more distantly related species a homolog should be "findable." If a protein's sequence similarity to its orthologs decays slowly, yet orthologs are not detected, this provides potential evidence for genetic novelty. The authors find that this is sometimes the case in two groups that have been well-studied for novel genes, yeast and insects. However, the authors show that it is much more common for orthologs to simply diverge so much that they become undetectable by BLASTP. This key result, and supporting analyses asking critical questions such as whether considerations of synteny can clarify matters, will be important for anyone working in this area to grapple with if they attempt to claim a gene is novel. Helpfully to the research community, the authors also include both the code used for the analysis and an easily searchable website where yeast and fly researchers can enter their favorite genes to see in which species that should be detectable.

I have no major issues with the manuscript as written, as both the text and the figures are clear, well-reasoned, and easy to follow. The authors are also upfront about some minor limitations to their analyses, which is appreciated. The minor points below are simply for the authors' consideration as potential additional details to consider discussing:

1. I have been trying to think through whether there are any limitations to using the BUSCO set of proteins to draw inferences about potential lineage-specific proteins, since these classes may differ in some notable parameters. For example, proteins in the BUSCO set are likely to be much longer than a hypothetical, recently evolved de novo evolved protein, to evolve more slowly (since wide conservation is a criterion for inclusion in BUSCO), to have more complex structures, and to perhaps be less prone to duplication. Do the authors think that any of these differences place any limitations on their analyses? I suspect not, since shorter lengths and faster evolution are likely to exacerbate the issue of non-detectability... which just supports the authors' conclusion that these are major limitations to making a claim about novel evolution.

2. On a related note, one issue that the manuscript highlights for me is the special difficulty posed by protein length in assessing lineage specificity. Even the shortest proteins shown in the examples of Figure 1 are long compared to most of the proteins discussed in studies of novel/de novo genes (e.g., fly eIF4B is 459 a.a.). The graphs in the figure illustrate that the shorter the length, the lower a protein starts on the similarity score y-axis, and thus the less room the score has to fall before it an ortholog of a given distance becomes undetectable at e = 0.001. If the authors see any use in adding a bit in the discussion about how protein length affects their results and conclusions (e.g., are shorter proteins more likely to fall into the non-detectable category?), it could be interesting, but I defer to their judgement here.

3. Did the authors look into how much changing the BLASTP e-value cut-off would affect their results?

4. Another criterion sometimes used to support the de novo emergence hypothesis is that the protein should lack structural similarity to other proteins, as would be expected for a protein that recently emerged from random sequence. This might be useful to consider here, since structural conservation tends to decay less quickly than primary sequence conservation. If there is a good way of assessing this, do proteins that fall into the "should be detectable, but aren't" category (i.e., probability values close to 1 in the top row of graphs in Figs. 4-5) have any major patterns of structural differences from proteins that fall into the "too divergent to be detected" category (probability values close to 0 in these figures)? (If it is easier to look into this specifically for, say, the 25 yeast genes described on p. 9 whose absence outside of the sensu stricto group is poorly explained by homology failure, that would be fine.)

5. While the web server for checking individual genes is already very useful, it would also be helpful for the authors to list in a supplemental table the supposedly lineage-specific individual S. cerevisiae and D. melanogaster genes that were used in the analyses for Figs. 4-5, along with their probabilities of detection at each outgroup species "t." This could make it easier for other groups to contribute to the further characterization the manuscript calls for, the genes whose lineage-specificity cannot be easily explained by homology detection failure could be prioritized.

Thanks to the authors and the PLOS Biology for the opportunity to review such a nice, interesting, and important manuscript!

Reviewer #2: 

The authors ask whether high evolutionary rates could explain the occurrence of orphans, i.e. genes that are only found in specific lineages. This is an old question and the authors address it by proposing a null-model for the evolutionary divergence of proteins and comparisons of yeast and Drosophila gene sets in the framework of this model. The paper touches on rather general concepts of evolution of proteins that have come in cycles in the past decades. The question of detectability by BLASTP has also been subject of discussion since quite some time. Accordingly, the issues addressed, as well as the conclusions, have been discussed before in various combinations and at various times. Of course, with today´s genomic data, the conclusions could potentially be better supported. But the curse of the genomic era is that algorithms and statistics dominate the outcome, without carefully looking at underlying assumptions and possible artefacts. The present paper is no exception in this respect. In addition, it is presented in an odd way, starting from the manuscript organization to the failure of providing the relevant data. The latter compromises proper reviewing and I therefore recommend rejection at this stage, simply because a full evaluation of the specific claims is not possible. But even if properly updated, it would likely better fit into a journal specialized on molecular evolution or algorithm development.

General:

The idea of a null model for the evolution of proteins with a constant substitution rate over time goes back to Zuckerkandl after doing the first globin sequence comparisons (if I remember right). It sparked the question of the possible existence of a molecular clock, which was extremely controversial for some time. The discussions resolved into the conclusion that there is apparently indeed a set of proteins that follow clock-like patterns, but that there are also others that behave rather differently, with lineage-specific changes in substitution rates. Further, it has become clear that whole lineages of taxa can show accelerated substitution rates. In molecular phylogeny studies, it had therefore become standard to test clock assumptions before using a gene for a phylogeny, since one would otherwise deal with artefacts. Hence, the assumption of the present paper that a constant decay rate, evenly spaced across a protein, would be a suitable null-model is an idealistic concept, which is only partially supported by the available data. There were therefore good reasons why previous papers on this question have taken more complex parameters into account (as cited). Why one would want to fall back behind these standards is not clear. 

In the present paper, a lineage specific acceleration has not been considered. Diptera (e.g. including Drosophila) show lineage specific acceleration rates, while beetles (e.g. including Tribolium) show particularly low rates (Savard et al. 2006, BMC Evol Biol. 25;6:7). Hence, rate calculations obtained from fly comparisons cannot be directly projected to other insects without correcting for this effect. The yeast dataset would also have to be checked for this potential problem. 

Second, it is also important to check whether there are even or uneven substitution rates along the length of a given gene, i.e. whether it includes a domain with high conservation in a background of low conservation (for example, most transcription factors and receptors fall into this class). Since BLAST requires only a small seed sequence for detecting a homologue, a short conserved domain is often sufficient to find it, even though the E-value may be pretty low, because of the divergence around it. This was the key argument of Alba and Castresana (ref 24) and it is rather unclear why the current authors throw it over board. The claim that it is better to make more simple assumptions may be appropriate in physics, but is seldomly correct in biology. The few examples they show in the paper cannot convince, since these are picked examples - while the primary data for all analyses are not provided. I expect that one could also pick different examples from them that would show a different pattern. Further, the use of BUSCO genes for calibration (without providing the relevant primary data) is rather one-sided, since this is a very select group of genes (and anyway usually used for other purposes, i.e. it is unclear why rate validation could be based on it).

Another problem that has been extensively discussed in the past is whether the routines for gene annotation generate biases on their own, which in turn bias conclusions drawn from them. There are actually many papers that show this now, starting from the insight that a filter on minimum ORF length does not make much sense, to the realization that quite a few coding regions include more than one functional ORF. Also, annotators consider an annotation as less reliable when no homologs are found in other species and tend therefore to remove them. Hence, when one relies on the annotated list of proteins only, especially on secondarily "curated" lists as it is done here, one loses many of the novel genes. Accordingly, suggesting general fraction-numbers for the relative detectability (or non-detectability) of novel genes does not seem appropriate when one has a biased dataset from start.

Yet another old discussion is the question of proper alignments to calculate substitution rates. In the early times when people have started to do this, there were clear recommendations that every alignment had to be manually inspected, that indels had to be treated in a consistent way (usually removed throughout the alignment together with some flanking regions) and that length changes needed to be considered etc. In fact, producing an appropriate alignment was a major achievement on its own. The present paper just uses a single algorithm (MUSCLE) and runs it under default parameters only, apparently without further manual inspection. However, it is well known that different parameters need to be used for different proteins, especially the gap penalty parameter can make a huge difference for the substitution rate calculation derived from such an automatic alignment.

If one removes the quantitative message from the paper, there is still a credible qualitative message, namely that a set of genes diverges beyond recognition because of fast evolution, while other genes evolve more slowly, yet show no matches in distant species. This is an important conclusion, but merely confirms what has been shown before (ref 5). These previous authors had basically asked the same question, came to the same qualitative conclusions and even the discussion is similar. 

Finally - and this has also been an active discussion in the past years - the authors fail to distinguish between novel genes and de novo evolved genes. The papers and reviews on de novo genes from the recent years make it very clear that a proof for de novo evolution can only come from comparisons between very closely related species, where synteny with ancestral non-coding sequences can be shown. The question of detectability by BLAST is not an issue in these cases, since the genomes to be compared should me >90% similar anyway. Hence, the present paper adds at best speculations to this discussion. The discussion on novelty is also problematic, since it is based on the assumption that two proteins that have diverged beyond recognizability would still have the same function since they are homologs (i.e. have a common evolutionary descent). But, for example, the fore limbs of all vertebrates are homologous and have nonetheless developed different lineage-specific functions. Homology alone does not imply function and this should apply also to proteins. Hence, the identification of remote homologues by itself does not yield insight into the evolutionary novelty question. 

Actually, it would be much more appropriate if this paper would be framed within the context of the evolutionary origin question of proteins that goes back to the famous 1977 paper of Francois Jacob on "Evolution and Tinkering". A main conclusion in the present paper seems to be that the majority of genes has remote homologs, which implies that they would have arisen already at the times of the origin of life, i.e. the authors would probably support the Jacob scenario. It is up to them how they want to discuss this in the light of proven cases for de novo evolution, but this is actually the real question that they are addressing in their study. 

Overall, with its simplified assumptions, this paper has more the character of an academic exercise than leading us a large step forward. It uses a idealistic concept of protein evolution, develops a corresponding algorithm for testing it and comes up with some interesting observations in detail. However, the overall conclusions are not really new and the conclusions on percentages of detectability are based on ignoring the necessity for parameterization, as well as a limited dataset that has excluded the more interesting genes for this question from start. 

Specific comments:

- the separation of figures from the text with a further separation of figure legends from the figures makes the paper very difficult to review unless one prints it out; this is a bit outdated

- no line numbers are included, which makes it difficult to list specific comments; hence, although I would have many, I do not add them here, since the paper would anyway require a major revision

Reviewer #3: 

I found this paper thoughtful and very well written, presenting a novel and useful new methodology, and am happy to suggest that it be accepted by plos biology with some minor revisions. I have just a few general comments and specific suggestions for the manuscript:

General comments:

1) It strikes me that readers may be surprised by how well the proposed neutral model, which simply accounts for length and divergence time/evolutionary rate, fits the data, without accounting for any other forces, e.g. selection to retain function, etc. Some discussion of neutral divergence and the evolutionary forces acting at the protein level could be included in the introduction or discussion. 

On a similar note, possibly outside the scope of this work, I wonder if different categories of genes, or genes general to many lineages as opposed to being lineage specific, fit the null model equally well?

2) I would like to see more discussion of the effect of differences in evolutionary rate over time, and over different parts of the protein- especially since the model fit for b is worse to the real gene-specific evolutionary rate. 

3) Results page 16- I find the gene ontology enrichment results difficult to interpret, given that there is no comparison to GO enrichment searches for genes that are not poorly explained by homology detection failure.

4) discussion- vakirlis et al find evidence that considerably fewer genes are missed due to homology detection failure than the authors. Can the authors expand on this discrepancy further? I found it quite surprising, especially since these Vakirlis et al. also use yeast and drosophila. 

Suggestions for clarity:

- Notation: the authors use a and b for their model parameters. These seem somewhat arbitrary choices, I suggest replacing them throughout: for example, replace 'a' with 'l', for length and 'b' with something like 'r', for rate, in order to help readers follow and distinguish between them.

- Notation formatting inconsistency: e is not constantly italicised throughout the manuscript.

- Fig 1: the legend could include an overview of what is meant by similarity score, for completeness. The dashed line also appears to be a different width in a than in b, the line in 1A should be made finer, so it is more clearly not at 0. 

- Figure 4 and 5- the authors could consider rearrangement? Following the text as currently written makes the figure layout quite confusing. Rearrange to appear in same order as referenced in text, and/or include slightly longer, more descriptive figure legends to explain the middle row as a kind of positive control, and bottom to more fully explain that the highlighting is for the lineage specificity of genes being included. The focal outgroup species name could also be highlighted in the phylogeny for additional clarity.

-Introduction page 9 - inclusion of large divergence in introductory paragraph- this is potentially confusing and could be clarified, because the meaningful distinction between this kind of divergence and genes for which we fail to find homologs is not obvious. If we can't find a homolog, is that sufficient divergence for the evolution of new function?

- Results paragraph beginning 'As both the sensu stricto yeasts and the drosophilid flies are relatively young lineages': the authors could more clearly lay out their methodology, of including genes specific to lineages with older divergence times in their analysis. The current language of 'testing two additional lineages', seems a bit too brief and may be confusing. 

- Method page 12 : 

'Proteins in the other yeast and insect species' - this makes it sound like insects were included in the yeast analysis, which is not the case?

'Proteins in the other yeast and insect species satisfying this reciprocal best hit criterion were considered orthologs of the S. cerevisiae and protein' - This is quite unclear, 'protein' meaning the protein in the other yeast proteomes? I think generally this methods paragraph could be made clearer- possibly by breaking this section up into two paragraphs, with one describing data collection and ortholog searches, and one describing the reciprocal best BLAST hits methodology. 

Reviewer #4: 

[identifies himself as Arne Elofsson]

In this paper, the authors propose a novel method to estimate the probability if no homologs in related species are missed because of rapid sequence evolution. This is then applied to argue that a number of (but not all) earlier proposed de novo created genes most likely are not de novo created. This is a very nice mathematical model and it provides some valuable insights into non optimal assumptions made in earlier papers. However, the results are not that surprising for most people as the problem of fast evolving genes have always been discussed in the context of de novo creation. Anyhow, I think this is a valuable method that should be published, in particular if it could be used to stop still appearing papers assuming that homology detection is very good and we can trace back all protein domain family relationships to Luca. 

Major:

One underlying assumption in this study is that an entire gene has a uniform evolutionary rate. This is certainly not a correct assumption, even it for normal genes might be an acceptable assumption. For globular proteins the variation in evolutionary rates the general trend for variation in evolutionary rate between sites is dominated by residues being buried or exposed. However, here the authors focus on genes that for some reason appears to evolve very fast and certainly they are quite different. Quite many of these proteins are intrinsically disordered and this (in addition to a faster evolutionary rate) means that the ration between insertions/deletions vs mutations is different and that the amino acid preferences are different. I would assume that taking site specific evolutionary rate into account would not shift the results significantly (likely a small number of the remaining potentially de novo created genes remains).

Reviewer #5:

[please also see downloadable Word doc]

Virtually all genomes contain genes that lack detectable homologues beyond a certain evolutionary distance. There is high interest in understanding the evolutionary mechanisms that underlie the emergence of these lineage-specific genes. In this manuscript, the authors propose to contrast two classes of evolutionary mechanisms: mechanisms that involve "novelty", such as de novo gene origination or rapid divergence following a duplication event; versus "uneventful" mechanisms, where a steady evolutionary rate eventually leads to the lack of detectability of ancient homologues. The authors present an analytical approach to identify lineage-specific genes that could be explained by a "null model" of homology detection failure due to uneventful evolution at a steady rate. The authors define a simple and elegant mathematical model based on such a scenario and apply it to two lineages, focused on S. cerevisiae and D. melanogaster. They find that potentially a majority of lineage-specific genes could be explained by homology detection failure. The manuscript is very clearly written and has the potential to be a solid contribution to our understanding of the origins of lineage-specific genes and to the wider field of evolutionary genomics. The method itself, freely available on github, could prove very useful for the research community.

Major concerns

The issue of homology detection failure is an important one and has been posed before in the literature. While the authors cite the relevant papers, the manuscript does not sufficiently address how this simulation approach differs from previous ones, why it is better, and to what extent it finds the same or different results. In particular, the simulation approach by Moyers and Zhang was based on a similar premise and was applied to the same two focus species. it would be important to fully consider the similarities and differences between the approaches and compare the results. 

Relatedly, the introduction is surprisingly short and does not paint a full picture of the state of the art in the field. Key studies are only mentioned in the results and in the discussion. The manuscript would be greatly improved in the introduction were more complete. What previous studies have addressed the same question? What did they find and how? What evidence is there, apart from absence of similarity, for de novo gene emergence? Furthermore, the dichotomy presented by the authors in the second and third paragraph can be misleading since rapidly evolving duplicates do have homologues, while de novo genes do not. It should be explained more clearly, and put in the context of previous literature which grouped genes with homologues together and contrasted them to de novo genes (eg, Vakirlis et al elife 2020).

Another issue with context concerns what is known about "new" genes. For instance, multiple reports have suggested that they have a higher evolutionary rate than other genes. This is a key piece of context that could drastically impact the validity of the model. Indeed, with the authors approach, such genes will very often, possibly almost always, be explained by the null hypothesis when they are in fact truly "novel". This issue is hard to overcome an should at minimum be acknowledged and discussed in detail as it appears a to be an important limitation.

The authors need to provide solid evidence that their model can truly distinguish between an evolutionary rate that is fast, but constant (null model), and an accelerated rate (novelty model). This would also be helpful to address the above.

The model that the authors propose makes an array of assumptions. For example, we understand that since the predicted bitscores correlate well with the real ones, it's safe to use them. However in the insect dataset there is plenty of variability and a peak seemingly at 0 (supp fig 1). This is important because it shows that, at least in the insects dataset, there are many cases where the model doesn't fit. This of course is expected; after all no model is perfect. But the authors should make sure that there isn't anything particular about these genes that makes them deviate from the model (and whether this could be relevant to lineage-specific genes). This caveat should also be appropriately discussed. The source data for each gene should be made available. Another assumption made is that evolutionary rates are stable across sites; but this is widely known not to be the case. This is another important limitation and should be acknowledged.

Synteny guided similarity searches are presented in two parts in the manuscript. First, for those genes for which homology detection failure is a very likely explanation. In this part, we read that out of the 126 S. cerevisiae genes considered, only 24, so 1/5 have an orthologous locus in at least one outgroup, based on YGOB. First, it should be clear whether that means simply that the syntenic region is identifiable or that there is also a gene present. Given the number of comparisons and the level of conserved synteny in yeasts the former would be surprising. If the latter, the number of cases where the orthologous region can be identified but no gene exists should be provided and commented upon. This is an important control for the authors' main claim: given that these genes are the best candidates for having simply diverged beyond detectability at the level of the clade in question, we would expect that for many of them a candidate homologue would be present in the predicted genomic region. If this isn't the case the authors should discuss why. Second, for those genes for which homology detection failure seems very unlikely. In this part, the out of lineage orthologues are not provided in a supplementary table, as is the case previously. The authors should make these data available. 

By design the analyses only focus on the most conserved subset of lineage-specific genes: those with sufficient homology to be able to fit model parameters. Moreover, yeast dubious ORFs are not included although they have been proposed to be enriched for true evolutionary novelty. This should be discussed as it is context for interpreting the main findings of the manuscript, which in several places reads as if it could provide an estimate of what proportion of lineage specific genes are truly novel. In fact, the manuscript estimates proportions based on a restricted search space (that which is conserved enough for the model to be applied). More careful attention to the writing of these conclusions, including in the title, is requested. Note that this limitation to the search space could potentially also explain why the estimates are higher than in previous literature such as Vakirlis et al. 2020, a question raised in the discussion.

Additional questions and minor concerns

- Supp. Tables 2 and 5 are the same file. The legend of Supp. Table 3 does not correspond to what the table actually is. The .eps format for supplementary figures is in general problematic. 

- Why include a bitscore data point of the alignment of the protein against itself? This is not justified in the manuscript. How does removing this affect the results?

- Why is the fungi dataset only half as big as the insect one? Since this is a substantial difference it should be appropriately justified. 

- Why were orthologues predicted using a method that the authors themselves admit is less than perfect, instead of using pre-computed ones from one of the many available resources of orthologues identified using more sophisticated approaches than RBH (https://questfororthologs.org/orthology_databases) ? 

- Some data did not appear to have been made available. For example, the lists of the lineage-specific genes at the various clade levels, the probabilities for individual genes etc. All such data should be made available.

- In the supp tables, species names should be italicized.

---

## [Decision Letter · Decision Letter 2]

23 Jul 2020

Dear Sean,

Thank you for submitting your revised Research Article entitled "Many but not all lineage-specific genes can be explained by homology detection failure" for publication in PLOS Biology. I have now obtained advice from the original reviewers and have discussed their comments with the Academic Editor. 

Based on the reviews, we will probably accept this manuscript for publication, assuming that you will modify the manuscript to address the remaining points raised by the reviewers. Please also make sure to address the data and other policy-related requests noted at the end of this email.

IMPORTANT:

a) You'll see that while reviewers #3 and #4 are now satisfied, reviewer #5 still has some residual requests, mostly pertaining to the relationship between your approach and that of Vakirlis et al. Regarding these points, the Academic Editor says, "I do think the textual revisions need to make clear the deficiencies in the method be dispassionately discussed relative to the previously published method. I do not agree that major point two requires a large scale analysis but I would be in favor of a few cherry picked examples where lowering the number of homologs allows us to see how the method does. In this respect I am contradicting the reviewers but again I think the point can be made illustratively."

b) Please attend to my Data Policy requests further down.

We expect to receive your revised manuscript within two weeks. Your revisions should address the specific points made by each reviewer. In addition to the remaining revisions and before we will be able to formally accept your manuscript and consider it "in press", we also need to ensure that your article conforms to our guidelines. A member of our team will be in touch shortly with a set of requests. As we can't proceed until these requirements are met, your swift response will help prevent delays to publication.

*Copyediting*

*Published Peer Review History*

*Early Version*

*Submitting Your Revision*

Sincerely,

Roli

Senior Editor

PLOS Biology

DATA POLICY:

Regardless of the method selected, please ensure that you provide the individual numerical values that underlie the summary data displayed in the following figure panels as they are essential for readers to assess your analysis and to reproduce it: Figs 1, 3, 4, 5 and all Supplementary Figs. NOTE: the numerical data provided should include all replicates AND the way in which the plotted mean and errors were derived (it should not present only the mean/average values).

REVIEWERS' COMMENTS: 

Reviewer #3: 

The authors' have addressed all of my original comments, and clarified a number of important points in the manuscript in response to the reviewers. The authors have also included additional analyses which further highlight the utility of their method. I have no further comments, and recommend that the manuscript be accepted for publication. 

Reviewer #4:

[identifies himself as Arne Elofsson]

My questions are fully satisfied and I think this paper is ready for publication,

Reviewer #5:

In this revised manuscript, the authors have clarified where the novelty of their work lies relative to the literature: providing a method to test whether a particular lineage-specific gene could be evolving under an evolutionary null model, independently from the syntenic conservation of that gene. The other points discussed in the revised manuscript (that blast based method are not sufficient to identify true evolutionary novelty, and that lineages-secific genes encompass some true novelty and some homology detection failure) are not novel at this date. Now that this is clarified, we agree that this novel method is a promising addition to the field, and would warrant publication if two major changes are made to the manuscript:

1 - in the writing of the manuscript (the whole manuscript, from title to discussion), be clear about the limitations of the method, specifically that it can only be applied to subset of genes: those with at least two known homologues within a lineage, and no known homologues outside of the lineage. In the two lineages studied, this is 155 annotated yeast genes (of 375 annotated lineage specific genes), 167 dubious yeast genes (of 784 dubious yeast genes), 1278 drosophila genes (of 1611 lineage specific drosophila genes). These numbers do not justify the generality of claims made by the authors of having applied their method to "all" lineage-specific genes, nor their quantitative and qualitative conclusions reflected in their title and discussion (including that "nearly all" dubious orfs fail to be detected).

2 - Given that the method, and its application to individual genes, are the key novel points of the manuscript, the authors need to demonstrate that the method indeed works on lineage specific genes. That is, they need to show that the parameter estimation step of their methods yields accurate results when only two homologues in closely related species are used. This can be done by using conserved genes and estimating the parameters when considering all versus only two closely related homologues, and comparing the results. This needs to be done at large scale, rather than on hand picked examples, in order to quantify the success rate of the method. Without such a quantitative evaluation of the method in the context it is meant to be applied to, we do not know if it really is informative.

Additional points:

(new point) Several of the modifications and new analyses presented by the authors seem to points to indels/gaps as a potential major limitation of the approach. It may be difficult to address for this manuscript, but it would be nice to point this out clearly in the discussion and present it as an exciting opportunity for future research in the field.

34. in summarizing the recent work of Vakirlis et al. the authors mention that the method Vakirlis et al applied does not allow to determine which particular lineage-specific genes may be due to homology detection failure. This statement is false. What is true is that that method cannot determine all of such genes across the genome (it only finds those with adequate synteny). But of course, neither does the authors' approach (it only applies to those with two or more homologues).

36. The issue is not adequately addressed in the discussion. The authors need to clarify that "it is possible for a gene to be consistent with the null hypothesis yet nonetheless be a novel lineage specific gene" if, as suggested by multiple reports, novel genes have high evolutionary rates. The discussion, as well as the title of the manuscript, read as if failure to reject the null hypothesis should be interpreted as the gene is not novel because that is the most conservative assumption. Instead, the percentage of genes that are truly novel but fail to reject the null hypothesis (ie, the false negative rate of the author's method) is unknown. This is a general limitation of the method and should be clearly stated as such.

43. This point was not sufficiently addressed in the author's response, so it is is reiterated as the first major comment here. However, the authors have added an additional analysis about Dubious ORFs which is interesting but presents some important limitation in the statistical analysis of the results and in their interpretation. 

- Do all genes analyzed have a detectable homologues in S Kud? If not, how can then all the distances be measured relative to S Kud? Then the authors state that "We chose S. kudriavzevii because it is the most distant species from cerevisiae according to our analysis (Figure 2)." But in Figure 2 the most distant Saccharomyces species from cerevisiae is S. bayanus. Is this a typo? Perhaps the methodological explanations can be better described here.

- The conclusions and interpretations are not statistically supported. All comparisons between distributions must be quantified with effect size and p-value in order to make any conclusion. 

- The authors make no mention of the evolutionary rate comparison within and outside of microsyntenic regions presented by Vakirlis et al. It is crucial, if the authors want to support their point, to do a fair comparison and discuss their results in the context of the Vakirlis results. The authors could show their distributions separately for genes also included in the Vakirlis et al. dataset and those ~400 not included, and do separate statistics for the two groups. This would immediately show if a bias is present in these genes specifically, or if it's a matter of the different measures being used for evolutionary rate. 

46. Our question was referring to the difference in number of species, not number of genes. This question still requires an explanation.

---

## [Editor Report · Decision Letter 3]

21 Sep 2020

Dear Dr Eddy,

On behalf of my colleagues and the Academic Editor, Harmit S Malik, I am pleased to inform you that we will be delighted to publish your Research Article in PLOS Biology. 

Early Version

PRESS 

Kind regards,

Alice Musson

Publishing Editor, 

PLOS Biology

on behalf of

Roland Roberts,

Senior Editor

PLOS Biology